# A systematic review and meta-analysis of the prevalence of childhood undernutrition in North Africa

**Nagwa Farag Elmighrabi** [1,2,3] *, **Catharine A. K. Fleming** [1,4], **Mansi Vijaybhai Dhami** [4,5], **Ali Ateia Elmabsout** [3], **Kingsley E. Agho** [1,4,6]

**1** School of Health Sciences, Western Sydney University, Campbelltown Campus, Penrith, NSW , Australia, **2** Organization of People of Determination and Sustainable Development, Libya, **3** Department of Nutrition, Faculty of Public Health, University of Benghazi, Benghazi, Libya, **4** Translational Health Research Institute (THRI), School of Medicine, Western Sydney University, Penrith, NSW, Australia, **5** Hunter New England Health, New Lambton, NSW, Australia, **6** Faculty of Health Sciences University of Johannesburg, Johannesburg, South Africa

* 20746360@student.westernsydney.edu.au

**Data Availability Statement:** All relevant data are within the paper and its Supporting Information files.

## Abstract

Undernutrition (stunting, wasting and underweight) among children remains a public health concern in North Africa, especially following recent conflicts in the region. Therefore, this paper systematically reviews and meta-analyses the prevalence of undernutrition among children under five in North Africa to determine whether efforts to reduce undernutrition are on track to achieving the Sustainable Development Goals (SDGs) by 2030. Eligible studies published between 1st January 2006 and 10th April 2022 were searched for, using five electronic bibliographic databases (Ovid MEDLINE, Web of Science, Embase (Ovid), ProQuest and CINAHL). The JBI critical appraisal tool was used, and a meta-analysis was conducted using the 'metaprop' command in STATA, to estimate the prevalence of each undernutrition indicator in the seven North African countries (Egypt, Sudan, Libya, Algeria, Tunisia, Morocco, and Western Sahara). Due to the significant heterogeneity among studies ($I^2$ >50%), a random effect model and sensitivity analysis were conducted to examine the effect of outliers. Out of 1592 initially identified, 27 met the selection criteria. The prevalence of stunting, wasting and being underweight were 23.5%, 7.9% and 12.9%, respectively. Significant variations between the countries with the highest rates of stunting and wasting were reported in Sudan (36%, 14.1%), Egypt (23.7%, 7.5%), Libya (23.1%, 5.9%), and Morocco (19.9%, 5.1%). Sudan also had the highest prevalence of underweight (24.6%), followed by Egypt (7%), Morocco (6.1%), and Libya (4.3%) with more than one in ten children in Algeria and Tunisia having stunted growth. In conclusion, undernutrition is widespread in the North African region, particularly in Sudan, Egypt, Libya, and Morocco, making it challenging to meet the SDGs by 2030. Nutrition monitoring and evaluation in these countries is highly recommended.

**Funding:** The authors received no specific funding for this work.

**Competing interests:** The authors have declared that no competing interests exist.

## Introduction

Malnutrition, whether under- or over-nutrition, affects hundreds of millions of children worldwide. Approximately, one hundred and forty nine (149.2) million, 45.4 million, 38.9 million and 12.6% of children under-five suffer from stunting, wasting, overweight and underweight, respectively [1]. Globally, undernutrition is responsible for approximately 45% of deaths among children under five [2]. To measure undernutrition, the three anthropometric indices, namely, stunting (low height-for-age), wasting (low weight-for-height), and underweight (low weight-for-age) are used. These three undernutrition indices also represent one of 100 indicators used to measure a community's progress toward achieving the 17 Sustainable Development Goals (SDGs) [3]. In this study, we focus on the three anthropometric indices of undernutrition (stunting, wasting, and underweight) in North African countries (Egypt, Sudan, Libya, Algeria, Tunisia, Morocco, and Western Sahara).

Nutrition during the first 2000 days (the first five years) of life is critical for the physical, cognitive, social, and emotional development of a child. Irreversible effects on a child's physical and mental health will occur if nutritional needs are not met during this period [2, 4, 5]. Yet, the burden of undernutrition remains high among children in North African countries. For instance, stunting affects more than 35% of under-five children in Libya and Sudan. At the same time, 22.3% of children suffer from stunting in Egypt, while 6.6% of children in the entire region suffer from wasting [1]. Undernutrition has profound direct and indirect effects on individuals and families due to a child's increased susceptibility to illnesses resulting in a heightened risk of morbidity and mortality [2]. Furthermore, undernutrition affects the economy and productivity of the entire community. Besides impairing physical growth, it also decreases academic performance, cognitive abilities, work efficiency, and earnings of individuals. Undernutrition may also result in increased costs of health care services, thereby causing the economy to grow slower, leading to less investment in human capital [6]. In Egypt and Sudan, the health costs of undernutrition among children amounts to approximately 1.10 billion Egyptian Pounds (EGP) and 11.66 billion Sudanese Pounds (SDG), respectively. In Egypt, 11% of child deaths are attributable to undernutrition and this is expected to increase by 32% by 2025, resulting in a cost of 26.8 billion EGP [7, 8].

To eliminate all forms of undernutrition, it is imperative to invest in interventions, monitor the progress of essential government initiatives and collect and analyse quality data [9]. In North African countries, there is a lack of current and specific information regarding undernutrition among children [10]. Previous studies on child undernutrition, carried out in various individual North African countries, namely Tunisia [11]; Egypt [12]; Sudan [13]; Algeria [14]; Morocco [15] and Western Sahara [16], highlighted the prevalence of undernutrition among children under five. Joulaei et al. [17]. conducted a review to examine the prevalence of stunting among children and adolescences aged 2–18 in 24 countries in the Middle Eastern and North African regions [17]. Apart from including a wide range of children ages with different growth requirements and health needs which affects the validity and generalizability of the results to all age groups, there was no information on Libya, Algeria, or Tunisia. A recent study in the Eastern Mediterranean region (including Afghanistan, Bahrain, Djibouti, Egypt, Iran, Iraq, Jordan, Kuwait, Lebanon, Libya, Morocco, Pakistan, Oman, Palestine, Qatar, Saudi, Somalia, Sudan, Syrian, Tunisia, Emirates and Yemen) found a prevalence of malnutrition [18]. These findings did not specifically target North African countries as the results combined the area with other regions or countries; this may result in an over- or under-estimation of the problem due to a separation from local contexts. More concerningly, this may result in the misrepresentation of data used to inform local policies, strategies and decisions [19]. The limitations of the study included an over-reliance on grey literature, no peer-reviewed journal

articles, and the limited timeframe of the search (2019–2020). Two years is too short a period of time to gather and analyse trends in the wider literature, or to reach comprehensive representative results as nutrition guidelines are typically updated every 5 years [18, 20, 21]. Finally, Algeria was not included in the review which could affect the generalizability of the findings to all Eastern Mediterranean countries including North African countries.

As discussed above, childhood undernutrition plays an increasingly significant role in childhood morbidity and mortality. Despite this, there are currently no ongoing nutrition monitoring and evaluation programs in North Africa to target the reduction of childhood undernutrition. Investigating how the prevalence of stunting, wasting, and underweight varies across North African countries would offer insights into how well these countries are progressing towards SDGs 1, 2, and 3 by the year 2030.

This systematic review and meta-analysis study addresses this gap. It was conducted to determine the prevalence of undernutrition (stunting, wasting, and underweight) among children under five in the North African region. The findings of this systematic review aim to help strengthen existing child nutrition policies and practices and enable policymakers to determine whether countries in the North African region are on track to meet SDGs 1, 2, and 3 by 2030. Results of the study are specific to the region and may lead to more efficient and practical applications of detailed nutrition interventions aimed at improving child health and nutrition in North Africa.

## Methods

### Anthropometric indicators

The current systematic review utilised three anthropometric indicators for analysis:

1. *Stunting* (chronic malnutrition): an indicator of retardation of linear growth and cumulative growth deficits in children; height-for-age <-2 standard deviation (SD) of the World Health Organisation (WHO) child growth standards median.

2. *Wasting* (acute malnutrition): measuring of body mass concerning the height and describing the current nutritional status; weight-for-height <-2 SD of the WHO child growth standards median.

3. *Underweight*: is a composite index of height-for-age and weight-for-height. It takes into account both acute malnutrition (wasting) and chronic malnutrition (stunting), but it does not distinguish between the two; weight-for-age <-2 standard deviations (SD) of the WHO child growth standards median [22].

### Search strategy

To review the existing literature, the study used the Preferred Reporting Items for Systematic Reviews and Meta-analyses (PRISMA) 2020 guidelines [23]. The PRISMA 2020 checklists for the manuscript and abstract are attached as additional S1 and S2 Tables. The protocol was submitted and registered with the Prospective Registry of Systematic Reviews (PROSPERO number CRD42022311922). Five databases were scanned for relevant peer-reviewed articles: Ovid MEDLINE, Web of Science, Embase (Ovid), ProQuest, and CINAHL. In addition, MeSH headings with related subheadings were combined with key terms for the included countries, region, and age-specific search terms were added to ensure all relevant studies were identified. In addition, the reference lists of retrieved articles and Google Scholar were screened for further relevant publications. Articles retrieved were imported from each database into an

EndNote library. The following combination of keywords was used in the search: (Child* or under-five* or paediatr* or infant* or bab*) AND (Malnutr* or malnourish* or undernourish* or undernutr* or stunt* or wast* or underweight*) AND (Egypt* or Sudan* or Libya* or Algeria* or Tunisia* or Morocco* or Western Sahara* or North Africa*) AND (prevalence* or rate*).

## Eligibility criteria

To be included in the review studies needed to meet the following criteria: 1) participants aged between 0 and 59 months of age; 2) conducted in any or all of the North African countries (Egypt, Sudan, Libya, Algeria, Tunisia, Morocco and Western Sahara); (3) investigated one or more anthropometric indicators of undernutrition (stunting, wasting, and underweight); 4) observational studies only (no qualitative, intervention, review, or experimental studies were considered); (5) published in peer-reviewed journals (no abstracts, policy briefs, or conference proceedings were reviewed); (6) written only in English and (7) published between 1 January 2006 and 10 April 2022. The year 2006 was used as a baseline in this review to capture the period when WHO Child Growth Standards were introduced by the WHO.

## Data extraction

All retrieved articles were imported from each database into EndNote X20 (Clarivate Analytics, USA) and sorted and ordering in a scientific and organised manner [24]. After removing duplicates, the first author, NFE, reviewed the study titles and screened all the publications. Following this initial screening, the second phase involved reading the abstracts and to identify the eligible studies. The third and final screening involved reading the full texts and ensuring each study met the eligibility criteria.

The identified studies are listed in Table 1, with the following information: The author, year of publication, geographic region, year of data collection, number of children and age, sample strategy, study design, population characteristics, and quality assessment score. In S4 Table, the included literature was classified according to the country of study and the prevalence of the three indices of undernutrition.

## Reliability

A second reviewer (MVD) who was blinded to the primary reviewer (NFE), checked the article's relevance and the recorded details. In each stage, the articles' appropriateness was checked, with any differences of opinion discussed. A third reviewer (KEA) was available to clarify differences regarding the final publications for inclusion.

## Quality assessment

The JBI critical appraisal tool was used to evaluate the degree of reliability, validity, and usefulness of the identified studies and to assess the quality of the reviewed studies. The tool is a set of 9 questions (a checklist) used to evaluate the external validity (bias in selection) and internal validity (information biases and bias of cofounders) [25]. The accuracy, suitability, and confidentiality of the JBI tool for critical appraisal of systematic reviews were confirmed [26]. For each reviewed study, a score ranging from zero to nine was allocated (zero if none of the criteria was met, and nine if all the criteria were met). This sum of points determined the overall quality of the study. Studies were rated as being high quality (7–9), medium (4–6) or poor quality (0–3), as shown in S3 Table.

**Table 1. Characteristics of included publications in the literature.**

| Author; year | Geographical Region | Year of Data Collection | Children (No); Age | Sample Strategy | Study Design | Population Characteristics | Quality Assessment Score |
|---|---|---|---|---|---|---|---|
| Abdalla et al. 2009 | El Fau (Rural area) in Sudan | 2003 | N = 150 6–60 months | Random sampling | Cross sectional study | Children under five in the three target villages | Medium |
| Abu-Manga et al. 2021 | Sudan | 2019 | N = 145,002 0–5 years | Multi-stage stratified sampling design | Cross sectional study | Children under five from households in the Sudanese community | High |
| Aitsi-Selmi 2014 | Egypt | 1992–95 | N = 9,201 0–5 years | Multi-stage stratified sampling design | Cross sectional study | Children under five in the community | High |
| | | 2005–8 | N = 13,376 0–5 years | | | | |
| Almasi et al. 2019 | Egypt | 2014 | N = 14898.9 0–5 years | Multi-stage stratified sampling design | Cross sectional study | Children under five in the Eastern Mediterranean countries | Medium |
| | Libya | 2007 | N = 10723 0–5 years | | | | |
| | Morocco | 2011 | N = 6872.6 0–5 years | | | | |
| | Sudan | 2014 | N = 11712.3 0–5 years | | | | |
| | Tunisia | 2012 | N = 2677.3 0–5 yrs. | | | | |
| Barouaca 2012 | Morocco | 2004 | N = 5311 0–5 yrs. | Multistage sampling surveys | Cross sectional study | Children under five in the community | Medium |
| | | 1997 | N = 5240 0–5 years | | | | |
| | | 1992 | N = 4502 0–5 years | | | | |
| Dahab et al 2020 | Sudan | 2014 | N = 14,081 0–5 years | Stratified multistage cluster sampling design | Cross sectional study | Children under five from households in armed conflicts in Darfur in Eastern state, Blue Nile and South Kordofan | High |
| Elsary 2017 | Egypt (Tamiya district in Fayoum) | 2014 | N = 400 0–5 years | Multi-stage stratified sampling design | Cross sectional study | Children under five who attend the health care facilities | High |
| El-Taguri et al. 2009 | Libya | 2003 | N = 7232 0–5 years | Multistage sampling designs | Cross sectional study | Children under five in five Arab countries (Libya, Djibouti, Syria, Morocco and Yemen | High |
| | Morocco | 2003 | N = 5380 0–5 years | | | | |
| El Taguri et al. 2008 | Libya | 1995 | N = 5348 0–5 years | Multi-stage stratified sampling design | Cross sectional study | Children under five from households across Libya | High |
| Fagbamigbe et al. 2020 | Egypt | 2014 | N = 13,682 0–5 years | Multi-stage stratified sampling design | Cross sectional study | Children under five in the community | High |
| Figueroa and Kurdi 2019 | Egypt | 2014 | N = 13601 6–59 months | Multi-stage stratified sampling design | Cross sectional study | Children under five in the community | Medium |
| | | 2008 | N = 9103 6–59 months | | | | |
| | | 2005 | 12131 6-59m | | | | |

(*Continued*)

**Table 1.** (Continued)

| Author; year | Geographical Region | Year of Data Collection | Children (No); Age | Sample Strategy | Study Design | Population Characteristics | Quality Assessment Score |
|---|---|---|---|---|---|---|---|
| Ghattas et al. 2020 | Egypt | 2014 | N = 13,857 6–59 months | Stratified cluster sampling | Cross sectional study | Children under five in Middle East and North African (MENA) region | High |
| | Algeria | 2013 | N = 13,077 6–59 months | | | | |
| | Sudan | 2014 | N = 12,538 6–59 months | | | | |
| | Morocco | 2003 | N = 5309 6–59 months | | | | |
| | Tunisia | 2012 | N = 2593 6–59 months | | | | |
| Kavle et al. 2015 | (Egypt) | 2008 | N = 6091 6–59 months | Multi-stage stratified sampling design | Cross sectional study | Children under five from households across Egypt | High |
| | | 2005 | N = 7794 6–59 months | | | | |
| Kerac et al 2019 | Egypt | 2014 | N = 1,210 0–6 months | Multi-stage stratified sampling design | Cross sectional study | Children less than six months of age in developing countries | High |
| Kiarie et al. 2021 | Sudan Yambio country | 2018 | N = 630 6–59 months | Cluster multistage sampling method | Cross Sectional study | Children aged 6–59 months living in Yambio County, Sudan | High |
| Mberu et al. 2016 | Egypt (Slum communities) | 2003 | N = 5761 0–5 years | Multi-stage stratified sampling design | Cross sectional study | Children under five in the community | Medium |
| Musa et al. 2014 | Sudan (Khartoum State) | 2014 | N = 411 0–5 years | Simple random sampling | Cross sectional study | Children under five in Khartoum state, Sudan | Medium |
| Nikooyeh et al. 2022 | Egypt | 2020 | N = 163 0–5 years | Online surveys | Cross sectional study | Children under five in Eastern Mediterranean countries | Medium |
| | | 2015 | N = 258 0–5 years | | | | |
| | | 2010 | N = 409 0–5 years | | | | |
| | | 2005 | N = 713 0–5 years | | | | |
| | | 2000 | N = 1142 0–5 years | | | | |
| | Sudan | 2020 | N = 29 0–5 years | | | | |
| | | 2015 | N = 99 0–5 years | | | | |
| | | 2010 | N = 266 0–5 years | | | | |
| | | 2005 | N = 80 0–5 years | | | | |
| | | 2000 | N = 28 0–5 years | | | | |

(*Continued*)

**Table 1.** (Continued)

| Author; year | Geographical Region | Year of Data Collection | Children (No); Age | Sample Strategy | Study Design | Population Characteristics | Quality Assessment Score |
|---|---|---|---|---|---|---|---|
| | Libya | 2020 | N = 74<br>0–5 years | | | | |
| | | 2015 | N = 111<br>0–5 years | | | | |
| | | 2010 | N = 225<br>0–5 years | | | | |
| | | 2005 | N = 320<br>0–5 years | | | | |
| | | 2000 | N = 241<br>0–5 years | | | | |
| | Morocco | 2020 | N = 286<br>0–5 years | | | | |
| | | 2015 | N = 347<br>0–5 years | | | | |
| | | 2010 | N = 640<br>0–5 years | | | | |
| | | 2005 | N = 652<br>0–5 years | | | | |
| | | 2000 | N = 679<br>0–5 years | | | | |
| | Tunisia | 2020 | N = 507<br>0–5 years | | | | |
| | | 2015 | N = 960<br>0–5 years | | | | |
| | | 2010 | N = 1064<br>0–5 years | | | | |
| | | 2005 | N = 1212<br>0–5 years | | | | |
| | | 2000 | N = 985<br>0–5 years | | | | |
| Özaltin et al. 2010 | Egypt | 2005–2008 | N = 33440<br>0–59 months | Multi-stage stratified sampling design | Cross sectional study | Children under five in Egypt and Morocco | High |
| | Morocco | 2003–04 | N = 5544<br>0–59 months | | | | |
| Rico et al. 2010 | Egypt | 2005 | N = 3242<br>6–59 months | Multi-stage stratified sampling design | Cross sectional study | Children under five in the community | High |
| Seedhom et al. 2014 | Egypt (El-Minia) | 2014 | N = 700<br>6–24 months | Non-random sampling | Cross sectional study | Children 6–24 months in Minia region, Egypt | Medium |
| Shaker-Berbari et al 2021 | Egypt | 2014 | N = 5100<br>6–23 months | Multi-stage stratified sampling design | Cross sectional study | Children under five in the Middle East | High |
| | Sudan | 2014 | N = 4009<br>6–23 months | | | | |
| Sharaf and Rashad 2016 | Egypt | 2014 | N = 12,997<br>0–5 years | Multi-stage stratified sampling design | Cross sectional study | Children under five in the community | Medium |
| Sulaiman et al. 2018 | North Sudan (Rural area) | 2014 | N = 1635<br>0–5 years | Cluster multistage sampling method | Cross Sectional study | Children under five who live in the Four Nile River rural area of Sudan | High |

(*Continued*)

**Table 1.** (Continued)

| Author; year | Geographical Region | Year of Data Collection | Children (No); Age | Sample Strategy | Study Design | Population Characteristics | Quality Assessment Score |
|---|---|---|---|---|---|---|---|
| Tzioumis et al. 2016 | Egypt | 2008 | N = 9275 0–5 years | Multi-stage stratified sampling design | Cross sectional study | Children under five in the community | High |
| | | 2005 | N = 12038 0–5 years | | | | |
| | | 2003 | N = 5911 0–5 years | | | | |
| | | 2000 | N = 10078 0–5 years | | | | |
| | | 1995 | N = 10165 0–5 years | | | | |
| | | 1992 | N = 7241 0–5 years | | | | |
| Winskill et al. 2021 | Egypt | 2014 | N = 11,706 0–5 years | Multi-stage stratified sampling design | Cross sectional study | Children under five in 39 low to middle-income countries | High |
| Zotarelli et al. 2007 | Egypt | 2000 | N = 10194 0–5 years | Multi-stage stratified sampling design | Cross sectional study | Children under five in the community | High |

High = 7–9; medium = 4–6, and poor = 0–3

## Statistical methods

The results were imported into STATA version 17 (Stata Corp LLC, Texas, USA) [27]. For each country, the prevalence of undernutrition indicators (stunting, wasting and underweight) was assessed through the forest plot. Each indicator was displayed in the forest plots with its corresponding weight, 95% confidence interval, and overall prevalence in the region.

A heterogeneity test of the different studies showed a high level of inconsistency ($I^2 >$ 50%), thereby indicating the use of a random effect model in all the meta-analyses conducted. Sensitivity analysis was also conducted by examining the effect of outliers. We employed a similar method to Patsopoulos et al. [28] which compared the pooled prevalence before and after eliminating one study at a time. The funnel plot was used to report potential bias and small/large study effects, and Begg's tests were used to assess asymmetry. Their respective funnel plots are shown as S1-S4 Figs.

## Results

As displayed in Fig 1, a total of 1592 peer-reviewed articles were initially identified from the five databases. After removing 519 duplicates, 1073 articles remained. The screening of the titles in the first screening phase excluded 986 articles. Further screening of the remaining 87 abstracts excluded another 37 articles. In the final phase, the full texts of the remaining 50 articles were screened and a further 30 articles were excluded. During the screening processes seven more papers were identified from Google scholar and other papers [29, 30]. Accordingly, 27 peer-reviewed papers were eligible for critical appraisal.

## Characteristics of included studies

As shown in Table 1, all 27 studies included in this review were cross-sectional surveys. Eighteen of these studies were investigated in Egypt [30–47], and 10 studies explored the problem in Sudan [32, 35, 38, 42, 48–53]. In addition, 6 studies assessed prevalence of undernutrition in Morocco [32, 35, 38, 39, 54, 55]; 4 studies assessed prevalence of undernutrition in Libya [32, 38, 55, 56], 3 studies demonstrated prevalence of undernutrition in Tunisia [32, 35, 38], and 1 study investigated in Algeria [35]. Six studies were conducted on children aged 6 to 60 months [30, 35, 40, 47, 48, 51], 2 studies investigated children aged 6–24 months [41, 42] and only 1 study investigated children from 0–6 months [36] and the remaining studies focused on children aged 0–60 months [31–34, 37–39, 43–47, 49, 50, 52–56]. Twenty two papers used secondary source data (nationally-representative data) [30–32, 34–40, 42–47, 49, 50, 53–56], and only 5 studies used a primary source data [33, 41, 48, 51, 52]. The prevalence of stunting was found in 22 studies [30–33, 35, 37–44, 46, 47, 49, 51–56], prevalence of wasting was found in 18 studies [32–34, 36–39, 41, 42, 45, 46, 48, 49, 51–53, 55, 56] and that of underweight in 12 studies [33, 37, 39, 41, 46, 49–54, 56].

Among the nine JBI criteria used to assess the quality of the studies, 18 (66.7%) articles scored seven points or higher [30, 31, 33–36, 39, 40, 42, 44–46, 49, 50, 51, 53, 55, 56], while 9 (33.3%) papers scored medium between 4 and 6 [32, 37, 38, 41, 43, 47, 48, 52, 54] as seen in S3 Table.

## Meta-analysis of the prevalence of undernutrition

**Stunting.** A forest plot of the prevalence of stunting among children under five in study is shown in Fig 2. The overall random effect pooled prevalence of stunting was approximately one in five, 23.5% with high heterogeneity ($I^2$ = 99.7%). Funnel plots and the use of Begg's test for stunting in North Africa indicated homogeneity (S1 Fig), and meta-regression analysis of undernutrition by year of publication indicated that the year of publication increased as the proportion of undernutrition decreased as shown in S4 Fig. However, this relationship did not differ statistically (p = 0.0083). Countries with the highest significant prevalence of stunting in the North Africa region were Sudan 36% (95% CI: 33.8, 38.2; P < 0.001), Egypt 23.7% (95% CI: 22.1, 25.2; P < 0.001), Libya 23.1% (95% CI: 21.0, 25.3; P < 0.001) and Morocco 19.9% (95% CI: 17.8, 22.0; P < 0.001). However, the overall pooled prevalence of stunting was not significant in Algeria 10.7% (95% CI: 10.17, 11.2; P < 0.001) and Tunisia 10.2% (95% CI: 9.1, 11.3; P < 0.001).

**Wasting.** Fig 3 presents a forest plot of the prevalence of wasting among children under five in the North African region. The overall random effect pooled prevalence of wasting was 7.9% (95% CI: 6.5, 9.3); p <0.001), with high heterogeneity ($I^2$ = 99.5%). The Funnel plots and use of Begg's test for wasting shown in S2 Fig reported the absence of publication biases. The highest significant rate of wasting in the region was reported in Sudan 14.1% (95% CI: 11.4,16.7; P < 0.001); and medium rates were found in Egypt 7.5% (95% CI: 6.1, 8.8; P < 0.001), Libya 5.9% (95% CI: 4.5, 7.2; P < 0.001) and Morocco 5.1% (95% CI: 2.1, 8.2; P < 0.001). However, the weighted prevalence in Tunisia was minor at 2.9% (95% CI: 2.5, 3.3; P < 0.001).

**Underweight.** The prevalence of underweight among children under five across the North African region, and in the individual countries under study is presented in Fig 4. Accordingly, the overall random effect pooled prevalence of underweight was 12.9% (95% CI:6.4, 19.5); p <0.001), with high heterogeneity ($I^2$ = 100.0%). The highest pooled prevalence of underweight was reported in Sudan 24. 6% (95% CI: 17.5, 31.6; P < 0.001) and the medium weighted rates of underweight were reported in Egypt 7% (95% CI: 5.1, 9.0; P < 0.001),

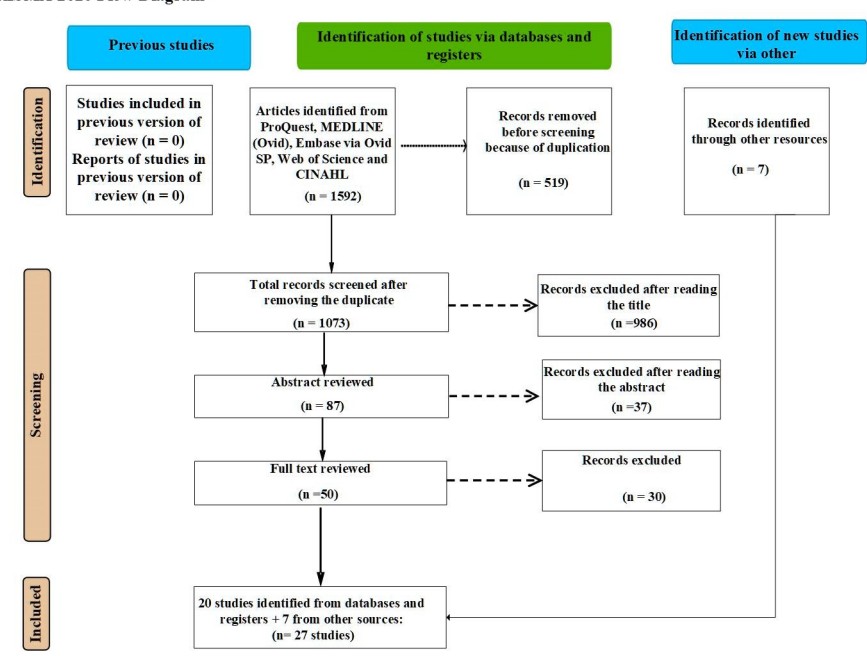

**Fig 1. Flow chart of the study selection process based on PRISMA 2020.**

Morocco 6.1% (95% CI: 3.4, 8.8: P < 0.001) and in Libya 4.3% (95% CI: 3.8, 4.9; P < 0.001). However, there was a little evidence of an effect on the prevalence, which implied homogeneity due to limited data. Funnel plots and the use of Begg's test for underweight, indicated an absence of publication biases, S3 Fig).

## Discussion

In the current study, 27 studies within the last 15 years met the eligibility criteria to determine the prevalence of stunting, wasting and underweight in North African children under five years of age. The critical appraisal of the 27 studies ranged from medium (9) to high (18). No studies were appraised as low. The overall prevalence of stunting, wasting and underweight in the current study were 23.5%, 7.9% and 12.9%, respectively.

Accordingly, the overall pooled prevalence of stunting among children under five in North Africa was approximately one in four. This is higher than that the global average of one in five. It is however lower than that of West Africa one in three children have stunting [57]. The review also found that the overall pooled prevalence of wasting among children under five in North Africa was approximately 7.9% which is higher than the global average of 6.7% and higher than the West Africa value of 6.9% [57]. Furthermore, the overall pooled prevalence of underweight among the children under five in North Africa was approximately one in eight. This is higher than the 2020 global average of 12.6% [58], and lower than the WHO 2018 estimate for the Africa region of 17.1% [59].

As described above, the study revealed stunting (chronic undernutrition) and underweight rates in children under five in North African countries were higher than the global average, while wasting was moderate (acute undernutrition). The results are consistent with a study by by Akombi et al [60]. Our results are also consistent with the study of Joulaei et al. [17] that focused on children 2–18 years in Middle East and North African countries, and found high

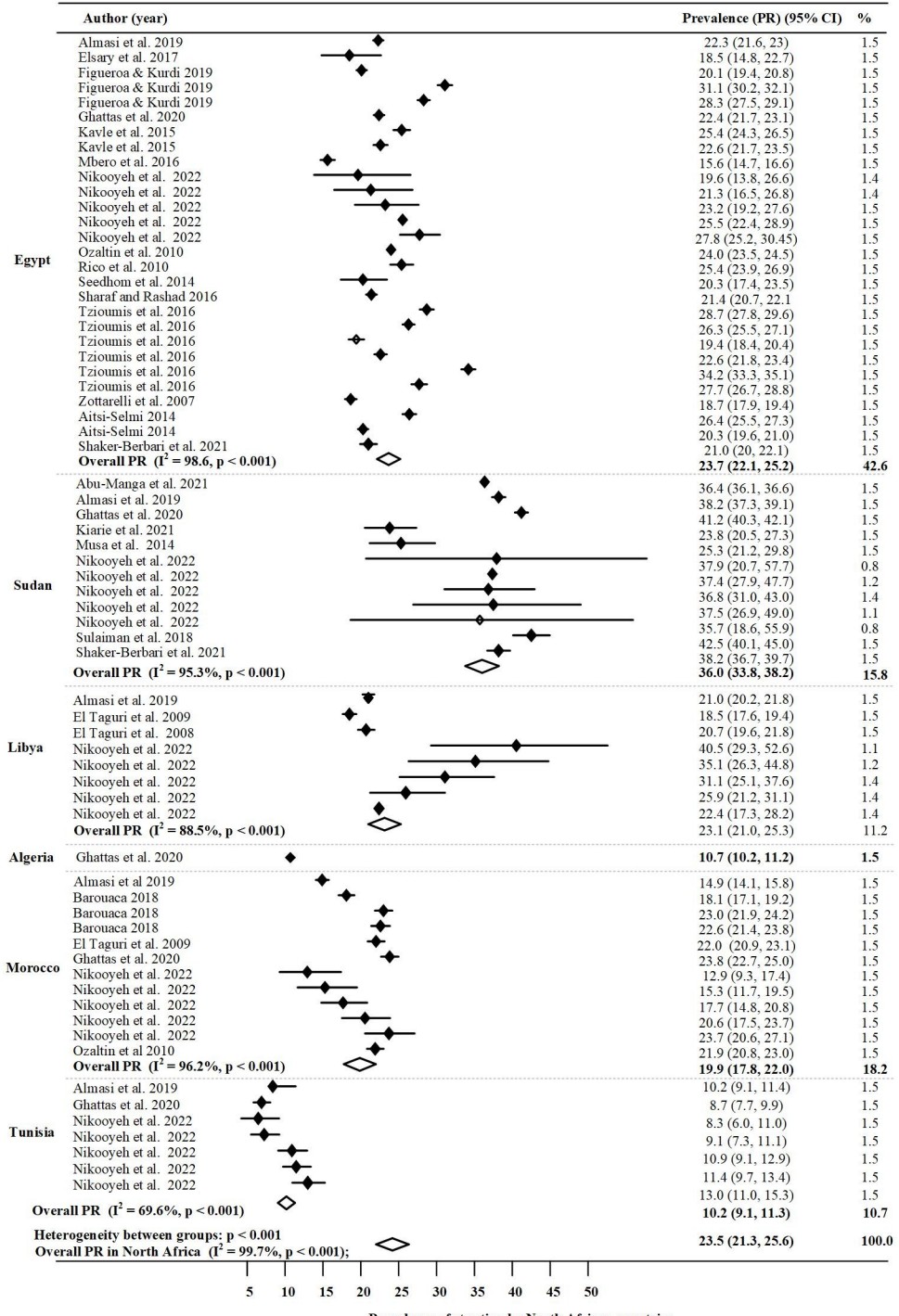

**Fig 2. Prevalence of stunting in North Africa by country.**

stunting rates among children [17]. Finally, our results agree with the 2021 WHO report which observed that stunting was prevalent and increasing in the North African region [1].

North African countries, similar to other low- and middle-income countries, is experiencing increased malnutrition as a result of conflicts and civil wars, increased food insecurity,

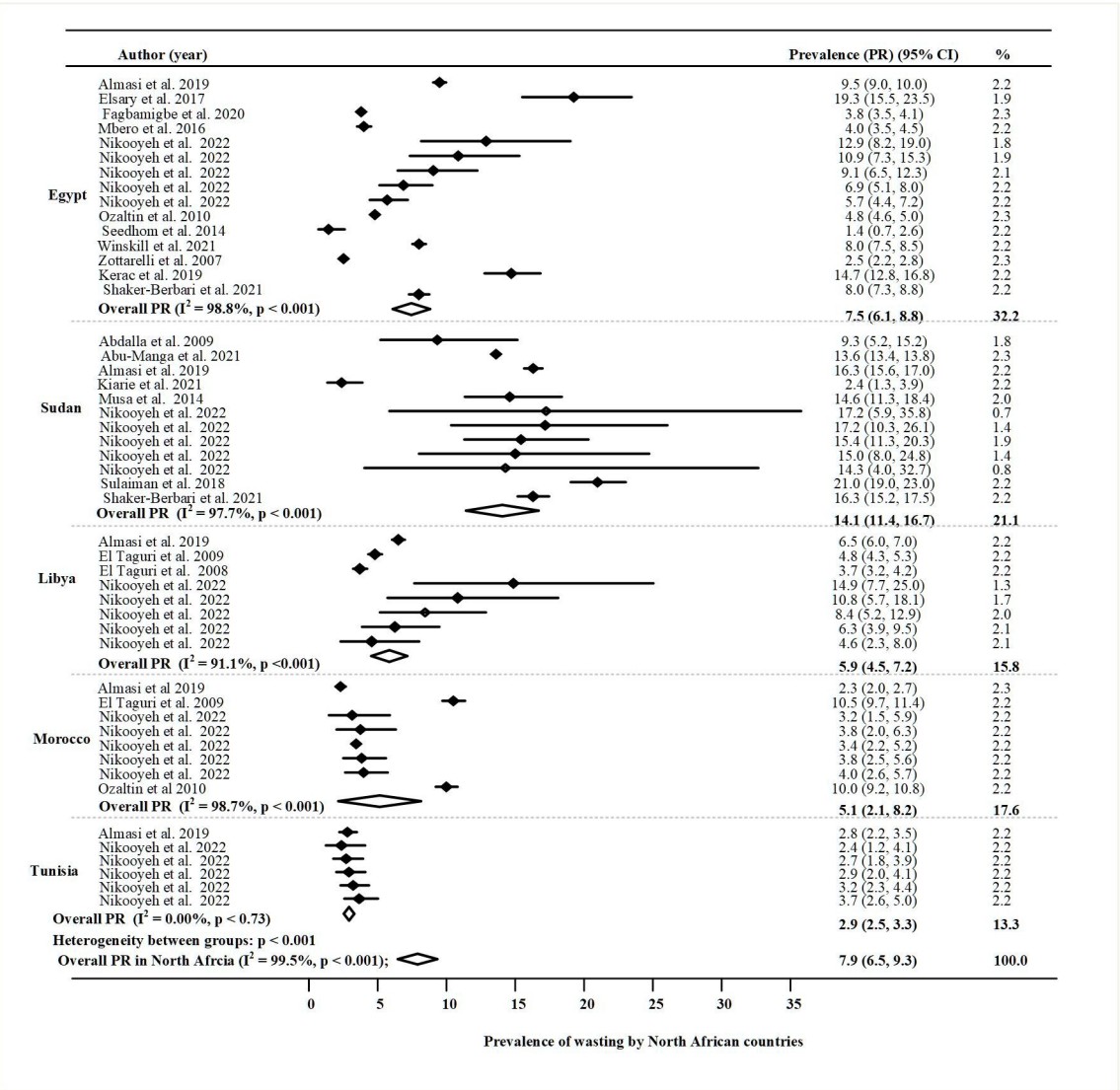

**Fig 3. Prevalence of wasting in North Africa by country.**

displacement, communicable diseases, and a changing climate [61–63]. For example, due to conflicts in Libya, children and families are experiencing inadequate dietary diversity and reduced access to food due to rapid declines in public services, particularly education and health services, as well as higher food and fuel prices, loss of shelter and livelihoods, loss of jobs and income [64]. Similarly, the ongoing crisis in Sudan has been exacerbated by a worsening economic situation, recurring violence throughout many states, a poor harvest, and global price shocks in grains and other food commodities. Children who live in conflict areas are often malnourished due to conflict displacement, infectious diseases, poverty, a decrease in food production, and a lack of essential resources such as food, water, and shelter. Most children aged 6 to 23 months did not consume the Minimum Acceptable Diet daily in Libya [64]. Additionally, families are unable to buy food at the market due to economic deterioration, political instability, and low salaries [65–68]. The effects of climate change are also possible contributors; Sudan and Libya, for example, are among the world's most water-scarce

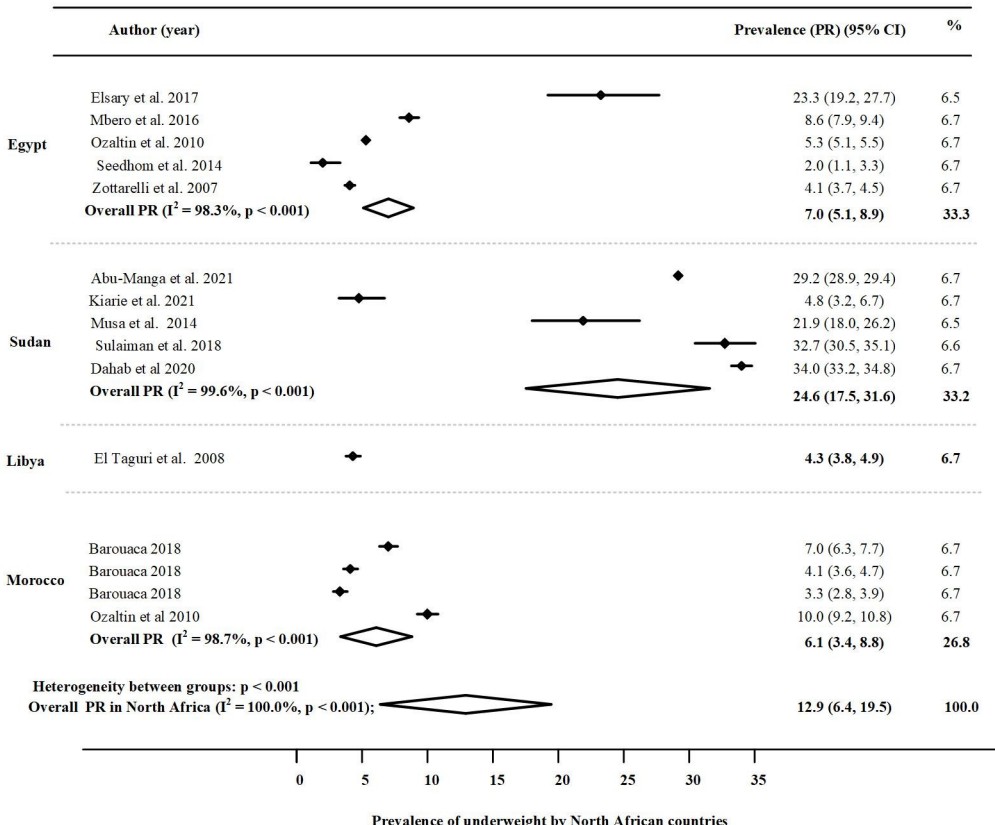

**Fig 4. Prevalence of underweight in North Africa by country.**

countries. Water, sanitation, and hygiene (WASH) services and facilities have been severely impacted by conflict, non-renewable water supply, and climate change in these countries [69]. Poverty might also play a role; 29.7% of Egyptians live below the official poverty line in 2019, despite Egypt having the third highest GDP in Africa [70, 71].

In the current study, we also found a significant variation between the prevalence rates of stunting and wasting among the countries with the highest rates, Sudan, and the lowest, Morocco. Sudan had the highest prevalence of underweight, followed by Egypt, Morocco, and Libya. More than one in ten children under five in Algeria and Tunisia had stunted growth. These conclusions align with several other studies. For instance, a study by Akombi et al. [60] on Sub-Saharan African countries found a substantial prevalence of undernutrition in the region, higher within East and West African countries than in South or Central countries in the same region [60].

The previously mentioned observations could be attributed to the concluded variation in the region, which also shows the high prevalence of undernutrition in countries like Sudan and Libya. That are still experiencing violent conflicts or countries like Egypt and Morocco, where conflicts have recently ended, compared with countries that have never experienced conflict or war.

The variation in undernutrition is likely due to high levels of disadvantage, inequality, and food insecurity in the North African region. This is exacerbated by conflict, displacement, and a changing climate. The United Nations Children's Fund (UNICEF) recently conducted a study of 11 countries in the Middle East and North Africa and found that poverty continues to

affect at least 29 million children, or one in four children in this region [72]. Poverty and unemployment rates, which are already high in Algeria, Tunisia, and Libya, are unevenly distributed across age, gender, and geography. Economic inequality is identified as a major obstacle to the global goal of ending global poverty by 2030 [73]. Africa represented the third unequal region after Latin America and Middle East [74]. The inequality may be attributed to economic, governmental, and social structures that limit the equal distribution of medical care, access to food, the impact on dietary intake, and other factors that affect child health directly or indirectly [75]. Poverty, food insecurity and inequality are the roots of undernutrition; these are exacerbated when regional conflicts and other environmental factors increase. For instance, studies from Bangladesh and South Asia found a strong direct, immediate association between higher income inequality and child health. The most vulnerable children suffer the most significant burden of stunting [76, 77]. Furthermore, in 2010, before recent conflicts begun, North African countries were already identified as food security-challenged countries, which faced challenges that prompted them to reconsider their approaches to development [78]. To achieve food security and development, it was suggested that these North African countries need to grow economically, increase employment, manage to sustain water resources, adapt to climate change in a timely manner [78].

## Policy implications of the study

Findings from our study may inform governments, policymakers, public health researchers and other stakeholders to identify countries in North Africa whose children are most vulnerable to undernutrition. The current study also serves as a needs assessment indicator for specific countries that report a high prevalence of child undernutrition. Our findings suggest specific policies are needed to address undernutrition within each North African country as an important public health issue. The current review contributes to the evidence for aid organizations and programming to continue allocating resources within North African countries to address undernutrition. This study also provides the latest update on national and international levels of undernutrition which can help determine whether North African countries are making progress towards achieving SDG goals 1, 2 and 3 by 2030.

## Strengths and limitations

The main strength of this review is its unique focus of undernutrition in the North African region, specifically in its seven countries of Egypt, Sudan, Libya, Algeria, Tunisia, Morocco, and Western Sahara. Other strengths include the use of uniform standards of measurement from the 2006 WHO standards for child health and the exclusion of studies in clinical settings (e.g., hospitalized children) thereby enhancing the generalizability of results to the general population. The study's eligibility criteria that evaluated the robustness of the study designs employed in each study was another strength. Nonetheless, this review has a certain number of limitations. Firstly, it was unable to investigate trends in the prevalence of undernutrition indices in the region due to inconsistent data collection and the limited number of studies from some included countries. Secondly, the selection of English-only studies likely biased the results towards studies in countries where the findings are reported in English. Nor did the current review examine the various factors that influence the epidemiology of undernutrition within this region.

## Conclusions

Undernutrition is a concern for children under five years of age in North Africa. Considerable variations by country were determined in undernutrition for stunting, wasting and

underweight. The North Africa region is at risk of an escalating burden of child undernutrition (particularly in Sudan, Libya, Egypt, and Morocco) due to ongoing conflicts, increased disadvantages and impacts of a changing climate. Public health attention must be paid to this region due to the wide range of inequalities experienced by families and young children living there. This review indicates that ongoing nutrition monitoring and evaluation programs are needed in North African countries to determine and measure progress made by these countries in achieving Sustainable Development Goals (SDGs) 1, 2 and 3 by 2030.

## Supporting information

**S1 Fig. Funnel plots and 95% confidence intervals of stunting.**
(TIF)

**S2 Fig. Funnel plots and 95% confidence intervals of wasting.**
(TIF)

**S3 Fig. Funnel plots and 95% confidence intervals of underweight.**
(TIF)

**S4 Fig. A meta-regression analysis of undernutrition by year of publication.** The vertical axis is the log proportion of undernutrition, and the horizontal axis represents the year of publication. Each dark dot represented one selected study, and the size of each dark dot corresponds to the weight assigned to each study. Given the slope of the regression line has descended slightly in this figure, this could be interpreted as the publication of year increased as the proportion of undernutrition decreased but this relationship differs statistically (p = 0.0083).
(TIF)

**S1 Table. PRISMA 2020 checklist for the main.**
(DOCX)

**S2 Table. PRISMA 2020 checklist for the abstract.**
(DOCX)

**S3 Table. Quality assessment score.**
(DOCX)

**S4 Table. Prevalence of undernutrition among children under five in North Africa.**
(DOCX)

## Acknowledgments

This paper is a part of the first author's doctoral dissertation at Western Sydney University's School of Health Sciences.

**Availability of data and other materials**

Sample size was not always explicitly stated in studies. The authors; however, direct readers to the name of the national data report or a link for more information. Demographic Health Survey (DHS) reports and databases for each country are available @ dhsprogram.com; information about Multiple Indicator Cluster Surveys (MIC) are available @ mics.unicef.org; and child malnutrition estimates data for 2021 can be found @ UNICEF/WHO/World Bank joint websites.

## Author Contributions

**Conceptualization:** Nagwa Farag Elmighrabi, Mansi Vijaybhai Dhami.

**Data curation:** Nagwa Farag Elmighrabi.

**Formal analysis:** Nagwa Farag Elmighrabi.

**Investigation:** Nagwa Farag Elmighrabi.

**Methodology:** Nagwa Farag Elmighrabi, Mansi Vijaybhai Dhami, Kingsley E Agho.

**Project administration:** Nagwa Farag Elmighrabi.

**Resources:** Nagwa Farag Elmighrabi.

**Software:** Nagwa Farag Elmighrabi, Kingsley E Agho.

**Supervision:** Kingsley E Agho.

**Validation:** Nagwa Farag Elmighrabi.

**Visualization:** Nagwa Farag Elmighrabi.

**Writing – original draft:** Nagwa Farag Elmighrabi.

**Writing – review & editing:** Nagwa Farag Elmighrabi, Catharine A. K Fleming, Ali Ateia Elmabsout, Kingsley E Agho.

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
