## [Decision Letter · Decision Letter 0]

5 Dec 2022

PONE-D-22-26443A systematic review and meta-analysis of the prevalence of childhood undernutrition in North AfricaPLOS ONE

Dear Dr. Elmighrabi,

Thank you for submitting your manuscript to PLOS ONE. After careful consideration, we feel that it has merit but does not fully meet PLOS ONE’s publication criteria as it currently stands. Therefore, we invite you to submit a revised version of the manuscript that addresses the points raised during the review process.

We look forward to receiving your revised manuscript.

Kind regards,

Seo Ah Hong, PhD

Academic Editor

PLOS ONE

Journal Requirements:

2. Please amend the manuscript submission data (via Edit Submission) to include authors Catharine A.K. Fleming, Mansi Vijaybhai Dham, Ali 

Ateia Elmabsout and Kingsley E. Agho.

3. Please include a copy of Table 2 which you refer to in your text on page 15.

5. We note that this manuscript is a systematic review or meta-analysis; our author guidelines therefore require that you use PRISMA guidance to help improve reporting quality of this type of study. Please upload copies of the completed PRISMA checklist as Supporting Information with a file name “PRISMA checklist”.

Reviewers' comments:

Reviewer's Responses to Questions

**Comments to the Author**

1. Is the manuscript technically sound, and do the data support the conclusions?

Reviewer #1: Yes

Reviewer #2: Yes

2. Has the statistical analysis been performed appropriately and rigorously? 

Reviewer #1: Yes

Reviewer #2: Yes

3. Have the authors made all data underlying the findings in their manuscript fully available?

Reviewer #1: Yes

Reviewer #2: Yes

4. Is the manuscript presented in an intelligible fashion and written in standard English?

Reviewer #1: Yes

Reviewer #2: Yes

5. Review Comments to the Author

Reviewer #1: The paper 'A systematic review and meta-analysis of the prevalence of childhood undernutrition in

North Africa' is written well, analysis is made in very good way. But need some minor correction, which are as follow.

1. Citation is not in correct manner.

2. Funnel plot if you made country-wide, heterogeneity can be easy assessed.

Reviewer #2: Well-done on your effort with this review of child undernutrition in North Africa. See below my comments to make your study stronger.

Abstract

Review this sentence in the abstract for grammatical error "Therefore, we aimed to systematically review and meta-analysis the prevalence of undernutrition .........".

What is the significance of this statement: "More than one in ten children in Algeria and Tunisia had stunted growth"?

Introduction

Overall: The introduction section appears disjointed with lots of grammatical errors which makes the narrative hard to comprehend. The authors attempt making some points but the points do not seem to flow logically across the paragraphs. I strongly advise a revision of the entire introduction section so it flows logically.

See below more specific comments.

In lines 65 - 66, what do you mean by "Burdens of undernutrition range from high to remarkably high among children in North African countries"?. How is "high to remarkably high" quantified? any reference? Please rephrase the sentence and include a reference.

Review this sentence in lines 69 - 70 for grammatical error: "It increases morbidity and mortality by exacerbates disease and makes children more susceptible to illness".

From lines 76 - 81, the dive into the discussion on health care cost was quite sudden and unrelated and does not add to nor continue the preceding discussion. It appears irrelevant to the on-going discussion.

Review this sentence in lines 81 - 82 for grammatical error: "Therefore, eliminating all forms of undernutrition is crucial, and requires investment interventions of collecting, analyzing, and quality data, which is the key....."

What do you mean in lines 92 - 95 by: "Apart from the child age, no studies were included on Libya, Algeria, Tunisia, or Western Sahara, even though they were a part of the study. Only four included studies focused on Egypt, two on Sudan, and one on Morocco". What study are you referring to here? This sentence is quite confusing because it does not follow a preceding line of thought and providing a reference is not enough.

In lines 98 - 99 you wrote: "However, this study was limited in several ways: First, the authors indicated that they conducted literature search from PubMed and google scholar, but only reported prevalence from published reports". How is this a limitation? Are reports not found in PubMed and google scholar?

In lines 99 - 100 you wrote: "Secondly the authors’ literature search was restricted to the period 2019-2020;". How is this a limitation?

In lines 100 - 101 you wrote: "third, Algeria was not captured, which implied that the study could not be applied to all the countries in the region". Does you review address this? Does your review capture all North African countries given the recent conflicts?

Review this sentence in lines 107 - 110 for grammatical error and break sentence in two: "The main goal of this study is to determine through a systematic review and meta-analysis of the prevalence of undernutrition (stunting, wasting, and underweight) among under-five children in North Africa and aimed to ascertain if those countries are on course to achieve the SDGs 1, 2, and 3 by the year 2030."

Method

What is the difference between lines 135 - 136: "Furthermore, Google Scholar and the reference lists of retrieved articles were screened for further relevant publications." and lines 140 - 143: "Furthermore, to avoid missing any additional relevant publications, we searched the bibliographical references of all retrieved articles that met the inclusion criteria, complemented by citation tracking by the use of Google Scholar". In my view, both sentences are communicating the same message. So I recommend you remove one of them.

Why were grey literature not included? Given the situation in some countries in North Africa, some government or non-governmental organizations may have some useful documents on child undernutrition in their individual countries.

Results

Include in Table 1 the prevalence of stunting, wasting and underweight as reported in the individual studies OR create a separate table that outlines the prevalence of stunting, wasting and underweight which informs the forest plots.

Discussion

In lines 274 and 275, replace the words "utilised" and "investigate" respectively with more suitable words.

Review this comment in lines 281 - 283: "Accordingly, the overall pooled prevalence of stunting among the under-5 children in North Africa was approximately one in every five. This is higher than that of the global average of 22%. It is however lower than that of West Africa (30.9%)". If you intend making comparison then use the same metrics. What % is "one in every five?".

In lines 370 - 371, how does excluding hospitalized children enhance the generalizability of results to the general population?

6. PLOS authors have the option to publish the peer review history of their article (what does this mean?). If published, this will include your full peer review and any attached files.

Reviewer #1: **Yes: **Jang Bahadur Prasad

Reviewer #2: No

---

## [Author Response · Author response to Decision Letter 0]

5 Jan 2023

All comments above raised by the journal editor and reviewers have been addressed in "Response to reviewers' file".

---

## [Decision Letter · Decision Letter 1]

2 Feb 2023

PONE-D-22-26443R1A systematic review and meta-analysis of the prevalence of childhood undernutrition in North AfricaPLOS ONE

Dear Dr. Elmighrabi,

Thank you for submitting your manuscript to PLOS ONE. After careful consideration, we feel that it has merit but does not fully meet PLOS ONE’s publication criteria as it currently stands. Therefore, we invite you to submit a revised version of the manuscript that addresses the points raised during the review process. 1. I still found some grammatical errors and typos although the reviewers previously mentioned.  English editing is necessary, both to improve grammar, readability, and reduce redundancy.

For example of “because of: (line 55)”, “development, (line 62)”; “Another a recent review (line 86)”; “(23,24); Fourthly,”; ”Table1 (line 168)”; “Lybia 4.3 (line 276)”; “in in (line 312)” and so on.

Break sentences to improve readability (Line 91-98)

2. check the reference format. For example, Reference #57.

We look forward to receiving your revised manuscript.

Kind regards,

Seo Ah Hong, PhD

Academic Editor

PLOS ONE

Journal Requirements:

Reviewers' comments:

Reviewer's Responses to Questions

**Comments to the Author**

1. If the authors have adequately addressed your comments raised in a previous round of review and you feel that this manuscript is now acceptable for publication, you may indicate that here to bypass the “Comments to the Author” section, enter your conflict of interest statement in the “Confidential to Editor” section, and submit your "Accept" recommendation.

Reviewer #1: All comments have been addressed

Reviewer #2: All comments have been addressed

2. Is the manuscript technically sound, and do the data support the conclusions?

Reviewer #1: Yes

Reviewer #2: Yes

3. Has the statistical analysis been performed appropriately and rigorously? 

Reviewer #1: Yes

Reviewer #2: Yes

4. Have the authors made all data underlying the findings in their manuscript fully available?

Reviewer #1: Yes

Reviewer #2: Yes

5. Is the manuscript presented in an intelligible fashion and written in standard English?

Reviewer #1: Yes

Reviewer #2: Yes

6. Review Comments to the Author

Reviewer #1: The paper entitled 'A systematic review and meta-analysis of the prevalence of childhood undernutrition in North Africa' have used appropriate analysis and written well.

Reviewer #2: (No Response)

7. PLOS authors have the option to publish the peer review history of their article (what does this mean?). If published, this will include your full peer review and any attached files.

Reviewer #1: **Yes: **Dr. Jang Bahadur Prasad

Reviewer #2: No

---

## [Author Response · Author response to Decision Letter 1]

11 Mar 2023

Response

1- because of: (line 55)”

Authors response: sorted out (line 54)

2- “Development, (line 62)”

Authors response: sorted out (line 64)

3- Another a recent review (line 86)”

Authors response: sorted out (line 82)

4- “(23,24); Fourthly,”

Authors response: sorted our (line 93)

5- ” Table1 (line 168)”

Authors response: sorted out (line 171)

6- “Lybia 4.3 (line 276)

Authors response: sorted out (line 254)

7- “in in (line 312)”

Authors response: sorted out (line 284)

8- so on

Authors response: We conducted a thorough English editing process to improve the paper’s readability and clarity.

9- Break sentences to improve readability (Line 91-98)

Authors response: sorted out (line 82-95)

10- check the reference format. For example, Reference #57

Authors response: sorted out (Reference 55); we thoroughly reviewed and checked all references.

---

## [Editor Report · Decision Letter 2]

15 Mar 2023

A systematic review and meta-analysis of the prevalence of childhood undernutrition in North Africa

PONE-D-22-26443R2

Dear Dr. Elmighrabi,

We’re pleased to inform you that your manuscript has been judged scientifically suitable for publication and will be formally accepted for publication once it meets all outstanding technical requirements.

Kind regards,

Seo Ah Hong, PhD

Academic Editor

PLOS ONE
---

## [Editor Report · Acceptance letter]

27 Mar 2023

PONE-D-22-26443R2 

A systematic review and meta-analysis of the prevalence of childhood undernutrition in North Africa 

Dear Dr. Elmighrabi:

I'm pleased to inform you that your manuscript has been deemed suitable for publication in PLOS ONE. Congratulations! Your manuscript is now with our production department. 

Kind regards, 

on behalf of

Prof. Seo Ah Hong 

Academic Editor

PLOS ONE